# The rational use of thromboprophylaxis therapy in hospitalized patients and the perspectives of health care providers in Northern Cyprus

**Syed Sikandar Shah**[1]*, **Abdikarim Abdi**[1], **Barçin Özcem**[2], **Bilgen Basgut**[1]

**1** Department of Clinical Pharmacy, Faculty of Pharmacy, Near East University, Nicosia, North Cyprus, Turkey, **2** Cardiac Surgeon, Near East University Hospital, Nicosia, North Cyprus, Turkey

* Sikandarshah850@gmail.com

## Abstract

### Background

Despite the presence of effective strategies and standard guidelines for the prevention of deep vein thrombosis (DVT), a considerable proportion of patients at risk of developing thromboembolism did not receive prophylaxis during hospitalization, while others received it irrationally, thus led to unwanted side effects.

### Aim

This study aimed to evaluate the current thromboprophylaxis practice and management of hospitalized patients at risk of developing DVT, along with the assessment of health care providers (HCPs) knowledge, and attitudes regarding DVT prevention.

### Methods

An observational study was conducted in the general wards of two leading tertiary university hospitals in Northern Cyprus in which patients from multiple clinics were enrolled to investigate the rational use of DVT prophylaxis using the Caprini risk assessment tool. Patients were also followed for possible complications two weeks post-hospitalization. A cross-sectional study followed to assess the knowledge and attitude of HCPs regarding DVT risks and prophylaxis.

### Results

Of the 180 patients enrolled, 47.7% were identified as irrationally managed, 52.3% were identified as rationally managed, 77.8% of patients were identified as having a high level of risk. Notably, Four of thirteen patients who received more thromboprophylaxis developed minor complications. Additionally, 73.3% of nurses had not received DVT education. Furthermore, more than 50% of physicians and nurses achieved a low knowledge score for DVT risks and prophylaxis.

**Data Availability Statement:** All relevant data are within the paper and its Supporting Information files.

**Funding:** The author(s) received no specific funding for this work.

**Competing interests:** The authors have declared that no competing interests exist.

## Conclusions

A high degree of irrationality in the administration of thromboprophylaxis therapy to hospitalized patients was observed. The overall scores for HCPs indicated insufficient knowledge of DVT risk assessments and prophylaxis.

## Introduction

Deep venous thrombosis (DVT) is still a life-threatening condition with significant mortality and morbidity [1]. It typically affects the deep veins of the leg or pelvis [2]. Venous thrombo-embolism (DVT and pulmonary embolism) is the most frequent preventable cause of death among hospitalized surgical patients [3]. Every year, approximately 2 million people experience deep venous thrombosis, and approximately 0.6 million of these patients experience a pulmonary embolism (PE). PE causes the death of approximately 0.2 million patients annually [4].

The American College of Chest Physicians (ACCP) indicates that all hospitalized patients have a minimum of one risk factor for venous thromboembolism and approximately 40% show 3 risk factors or more [5], thus requiring adequate thromboprophylaxis to decrease mortality and morbidity [6]. Primary prophylaxis is the most common method and uses medications and mechanical methods to prevent DVT. Meanwhile, secondary prophylaxis is less commonly used and includes screening methods and the treatment of subclinical DVT [7]. Factors influencing the determination of appropriate prophylaxis include patient factors, setting, drug therapy, and knowledge of these aids in the accurate control of DVT. Evidence-based risk assessment tools (RAT) have been adopted to accurately evaluate these risk factors [8]. After a risk assessment, pharmacological prophylaxis regimens should be prescribed for moderate- to high-risk patients, while pharmacological prophylaxis may not be necessary for low-risk patients after a risk-benefit evaluation [9]. Irrational use of medications may lead to adverse drug reactions, waste of rare health resources, and increased treatment costs [10]. Many observers reported that healthcare providers may under or overestimate thrombosis risk factors in hospitalized patients, leading to either DVT or overmedication, which may result in bleeding and unwanted side effects [11].

The determination of competence of health care providers in deep venous thrombosis risk assessments and preventive measures may be valuable in improving their education and awareness and attenuating this significant public health issue. Multidisciplinary teams including clinical pharmacists, nurses, and physicians are needed to ensure rational drug use and adherence to evidence-based guidelines [12]. However, no study has assessed the rational use of DVT prophylaxis in tertiary care hospitals in North Cyprus.

This study aims to evaluate the current thromboprophylaxis practice and management of hospitalized patients having risks of developing DVT, along with the assessment of health care providers (HCPs) knowledge, and attitudes regarding DVT prophylaxis.

## Materials and methods

### Study setting and subjects

The study was conducted in the general wards of two tertiary university hospitals, NEU hospital in Nicosia and KUH in Kyrenia of Northern Cyprus. In the first phase, an observational prospective study was performed. All (n = 310) patients admitted between 01 April 2018 and

01 July 2018 who met the inclusion criteria were invited to participate in the analysis. The inclusion criteria were acute and chronically ill hospitalized patients for whom complete medical records were available and who were hospitalized for at least 7 days in a certain ward. Patients having age <18 years, superficial vein thrombosis, or any contraindications for DVT prophylaxis and patients who had deep venous thrombosis prophylaxis within the last month were excluded from the analysis.

Information was collected from eligible patients, who were assessed for risk factors and the rational use of prophylaxis for DVT using an evidence based DVT risk assessment tool.

Demographic information of patients willing to participate in the study were recorded including age, sex, height, weight, primary diagnosis, chief complaints. Also the presence of risk factors of DVT, a drug used for DVT, sign, and symptoms of DVT, laboratory results, other comorbidities, any prophylaxis treatment administered for VTE, and a history of signs and symptoms of PE or DVT or anticoagulant complications documented in the patient files during hospitalization were collected. Patients were also assessed for possible complications by the research team during their follow-up visit two weeks after hospitalization to record any deep venous thrombosis signs and symptoms, pulmonary embolism, or adverse effects of medications.

In the second phase performed between 5th September 2018 and 5th November 2018, a cross-sectional questionnaire was distributed to health care providers at the two health care settings in a face to face meeting to assess the knowledge, practices, and attitudes of health care providers towards DVT risks and prophylaxis.

## Study tools

**Risk assessment tool.**   The Caprini tool is a validated DVT risk assessment tool [13] that has been used in many healthcare settings worldwide to analyse hospitalized patients [14] and includes 20 variables [15].

The Caprini risk score for the assessment of thrombosis risk in adult hospitalized patients was used to categorize patient risk and accordingly identify the required thromboprophylaxis mode. Patients' risk factors are classified into four categories: "very low risk" (0 points), "low risk" (1–2 points), "moderate risk" (3–4 points), and "highest risk" (≥5 points).

**Health care providers questionnaire.**   Two different questionnaires were used to assess the knowledge, practice, and attitudes towards DVT. Questionnaires comprising 53 items for nurses [16] and 21 items for physicians [17] were adapted based on a literature review. The adapted questionnaire was reviewed by a committee of experts comprising a clinical pharmacist, pharmacologist, and cardiologist practicing in Northern Cyprus.

The first part of the questionnaire designed for nurses collects information about demographic characteristics using 12 questions. The second part comprises 20 questions assessing the nurses' knowledge of deep venous thrombosis risks with 3 choices (false, true and do not know), and the third part examines knowledge of the prevention of deep venous thrombosis using 8 questions with 3 choices (false, true and do not know). Both false and do not know responses were considered negative in the analysis. The fourth section examining the practices of nurses in deep venous thrombosis prevention consisted of 13 questions with a 3-point Likert scale (always, sometimes, and never).

A short questionnaire lacking demographic characteristics was distributed to physicians to increase the response rate. The adopted questionnaire consisted of two sections. The first section contained 15 questions, of which 11 questions assessed knowledge of DVT with 4 multiple choice responses while the other 4 questions had 2 choices (true and false). The second section examining the attitudes of physicians towards DVT prevention consisted of 6 questions with 5

choices (Strongly disagree, Disagree, Neutral, Agree, and Strongly Agree).Physicians' knowledge and attitudes were assessed using a questionnaire that included 15 knowledge-related questions scored from 0–15 points and 6 attitude-related questions scored from 6–30 points with a Likert scale. For the present study, favourable knowledge and attitudes were defined as a score greater than 70% [18]. Two native Turkish speakers with experience in translating health questionnaires independently translated the questionnaire. The two translators then compared their translations and a third questionnaire was produced jointly.

## Pilot study

A pilot study was performed that targeted 10 to 15% of the study population, i.e. patients (n = 35), nurses (n = 40) and physicians (n = 15) [19]. The internal consistency was measured for different scales using Cronbach's alpha and Kuder-Richardson (KR-21), which reflect good internal consistency (0.8) for both nurses' and physicians' knowledge and (0.7) for the attitudes of physicians.

## Ethical consideration

The study protocol was approved on 29th March 2018 by the Institutional Review Board (IRB) of Near East University (YDU/2018/56-530) and assigned as an observational study. A written consent form was signed by healthcare providers upon their participation in the study. Verbal consent was obtained from patients and recorded on data collection form upon their follow-up interview.

## Statistical analysis

Statistical Package for Social Sciences (SPSS) software, version 22.0, IBM corp., New York, USA was used to analyse the data. Descriptive statistics for qualitative and quantitative variables were used to analyse the results of the study. Categorical data are reported as frequencies and percentages (%), while continuous data are reported as the means (± standard deviations) or medians (ranges).

Raosoft software version 2.3 (Raosoft. Inc., Seattle, USA) was used to calculate the minimum sample size required for the study. Assuming a 95% confidence level, a 5% margin of error, and a 50% response distribution, at least 172 patients were needed to participate in the study out of 310 admitted to the hospital during the study duration. While 98 physicians and 169 nurses were required as a minimum required sample out of 130 physicians and 300 nurses providing care at the two hospitals involved in the study.

Following the testing of normality, non-parametric hypothesis tests were performed throughout the whole data analysis phase. The Mann-Whitney U test and the Kruskal-Wallis test were performed to compare data between multiple groups. The associations between categorical variables were analysed using Fisher's exact test and Pearson's Chi-square test. The level of significance was set to $P < 0.05$.

# Observational results

## Patient demographics and characteristics

One hundred eighty patients with multiple pathologies from the general wards were enrolled to investigate their risk of thrombosis. The mean age ± SD of the patients was 65.47± 16.39 years, and 59.4% were male and 40.6% were females. The median length of hospitalization stay was 15 with 29.75–7.00 IQR. The minimum number of risk factors for patients was 0 and the maximum number of risk factors was 14 (median of 6/patient). The most common drug used

**Table 1. Main demographic and clinical characteristics of the 180 patients N (%).**

| Clinics | Cardiology | Pulmonary | GIT |
|---|---|---|---|
| Number | 83 (46.1%) | 11 (6.1%) | 18 (10%) |
| | **DM** | **Orthopaedics** | **Neurology** |
| | 3 (1.7%) | 16(8.9%) | 22(12.2%) |
| | **Respiratory** | **Allergy and chest disease** | **Infectious disease** |
| | 5 (2.8%) | 2(1.1%) | 5 (2.8%) |
| | **Geriatrics** | **Oncology** | **Surgery** |
| | 7 (3.9%) | 7(3.9%) | 1 (6%) |
| Average age | 65.47 ± 16.39 (mean ± SD) | | |
| The average number of drugs | 9.41 ± 4.7 (mean ± SD) | | |
| Males | 107 (59.4%) | | |
| Females | 73 (40.6%) | | |
| High level of risk | 140 (77.8%) | | |
| Moderate level of risk | 27 (15%) | | |
| Low level of risk | 10 (5.6%) | | |
| Very low level of risk | 3 (1.7%) | | |
| Rationally managed cases | 94 (52.3%) | | |
| İrrationally managed cases | 86 (47.7%) | | |
| Patients with no need for prophylaxis (total) | 3 (1.7%) | | |

for thromboprophylaxis in patients was enoxaparin (58.8%). Notably, 4.4% of patients died during follow-up but the cause of death was not related to DVT. Table 1 presents the main demographic and clinical characteristics of the patients included in the present study.

The most common risk factors identified in the sampled patients included age of 41–60 years (26.1%), obesity (BMI>25) (21.1%), patients who were confined to bed for more than 3 days (100%), an age of 61–74 years (37.8%), and an age ≥75 years (28.3%). The distribution of risk assessment items and risk factors among sampled patients is shown in Table 2.

## Thromboprophylaxis and rationality

Of the 180 patients, thromboprophylaxis was appropriately provided to only 94 patients who received rational thromboprophylaxis. Of the 86 irrationally managed patients, 65 patients did

**Table 2. Distribution of the most common risk factors among sampled patients.**

| Risk factor | N (%) |
|---|---|
| Age of 41–60 years | 47 (26.1%) |
| Swollen legs | 16 (8.9%) |
| Obesity (BMI >25) | 38 (21.1%) |
| Serious lung disease, including pneumonia | 12 (6.7%) |
| Acute myocardial infarction | 8 (4.4%) |
| Congestive heart failure | 8 (4.4%) |
| Abnormal pulmonary functions (COPD) | 11 (6.1%) |
| Age of 61–74 years | 68 (37.8%) |
| Patient confined to bed for > 72 hours | 180 (100%) |
| Major surgery > 45 minutes | 11 (6.1%) |
| Minor surgery | 13 (7.2%) |
| Aged 75 or older | 51 (28.3%) |

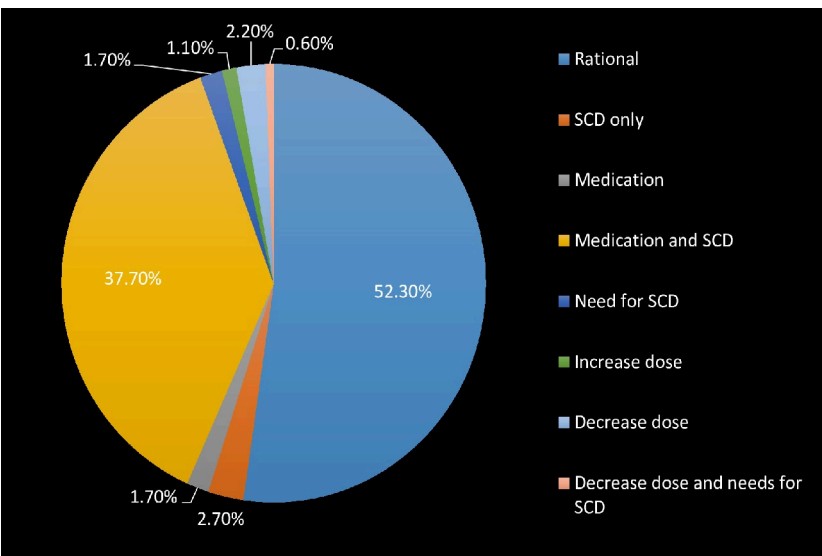

**Fig 1. Proposed management of the sampled patients based on the Caprini score.** SCD, Sequential Compression Device.

not take any form of thromboprophylaxis and 3 patients received inadequate prophylaxis (e.g., insufficient doses of enoxaparin or compression stockings alone). Thirteen patients received more thromboprophylaxis than was indicated (either taking an increased dose or taking medicine when only compression stockings were indicated). The only four of these 13 patients developed minor complications while anticoagulation therapy was stopped in 2 patients. The most common minor complications were wound haematoma, injection site bruising and haematuria. These minor complications developed mostly in elderly patients (>70). However, a statistically significant difference in complications was not observed between genders. No major complications (e.g. gastrointestinal or retroperitoneal bleeding, thrombocytopenia, or fatal pulmonary emboli) were recorded during hospitalization and post-hospitalization follow-up visits.

Fifty-eight patients out of 140 high-risk patients (41.4%) were not treated with thromboprophylaxis requiring both compression devices and an antithrombotic agent. Eight of these patients developed signs and symptoms of DVT (e.g. Warm feelings of legs in 4 patients, Leg swelling in 3 patients, etc.) Fig 1 shows the proposed management of the sampled patients based on the Caprini score.

Based on the data, 80.7% (n = 113) of the female patients and 75.7% (n = 106) of the male patients had high-risk factors, but no statistically significant associations were observed between gender and the categories of risk factors. Of the 104 patients aged greater than 65 years, 2.8% (n = 3) displayed a low level of risk, 7.6% (n = 8) of these patients belonged to the moderate risk group, and the other 89.4% (n = 93) were assigned the high-risk group. The presence of stroke, multiple trauma or acute spinal cord injury less than one month prior to DVT, hip or leg fracture, a family or personal history of VTE, hospitalization or treatment for cancer in the last year, and current immobility were among the minor risk factors and were the strongest independent predictors of VTE among sampled patients.

## Responses and characteristics of the nurses

Two hundred sixty-five questionnaires were dispensed to nurses, and 237 were returned, corresponding to a response rate of approximately 89.4%. 232 questionnaires were evaluated,

while 5 were improperly filled and discarded. Most of the respondents had a bachelor's degree (58.6%), were females (69%) and (53.4%) had <5 years of experience. Most of the respondents were working in internal medicine (16.8%), and emergency units (15.9%). The most common age group was <25 years (53.4%). Nurses' responses to the question "Did you receive previous education on deep venous thrombosis?" indicated that 73.3% of the respondents had not received DVT training. Those nurses (n = 62) who received DVT training reported 5 resources. Most of the nurses (n = 42) and (n = 9) had received this training at their congress/conferences and vocational high school, respectively. Other training resources were internet resources (n = 4), courses (n = 2) and workplace training (n = 5). Approximately (n = 206) of the nurses expressed that they needed education on DVT. Participants rated the quality of previous deep venous thrombosis education as excellent (n = 4), very good (n = 15), good (n = 31) and poor (n = 12).

## Nurses' knowledge of and practice in thromboprophylaxis

Most of the respondents recorded correct answers for most of the questions (6 of 6 questions) examining their general knowledge of DVT. They recorded correct answers for the statements "DVT occurs as a result of injury to a vessel wall, altered blood coagulation, and stasis of blood" (84.5%), and "DVT typically occurs in the lower extremities (deep leg veins)" (53.4%).

Most of the nurses had a low percentage of correct answers to most of the questions (5 of 8 items) examining their general knowledge of the prevention of deep venous thrombosis. They also recorded correct answers for the question "Exercise of the leg and foot (lower extremities) may prevent deep venous thrombosis" (77.2%). Furthermore, most of the nurses had a high percentage of incorrect answers to the question "Development of deep venous thrombosis may be prevented by elastic compression stockings."

The analysis of respondents' knowledge of deep venous thrombosis risk factors revealed a low percentage of correct answers to most of the questions (12 of 20 questions). The most common correct answers were recorded for the question "Prolonged immobilization may cause deep venous thrombosis in hospitalized patients" (78.4%), and the most common incorrect answers were recorded for the question "Inflammation or infections may predispose a patient to deep venous thrombosis" (71.6%).

Regarding the practice of nurses in preventing DVT, the investigation revealed that most of the participants responded with the option "always" to all questions compared with the choices "sometimes" and "never". The most common answers receiving a rating of "always" were recorded for the question "Educating the patients to avoid injury" (72.8%). The nurses more frequently responded with the choice "sometimes" to the question "Educating the patients about the appropriate utilization of graduated compression stockings" (29.3%) and frequently responded, "never" to the question "Educating the patients about adequate or sufficient fluid intake" (23.3%).

No statistically significant differences were observed in the four different scores between genders (p>0.05), as shown in Table 3. The median for practice on DVT prevention for nurses >31 years old was significantly lower than the median of nurses aged from 26–30 years and <25 years, (17), (18) and (21) (p<0.05), respectively. Meanwhile, the median for the general knowledge of DVT attained by nurses >31 years old was significantly higher than the median of nurses aged from 26–30 years and < 25 years old, (5), (4) and (4) (p<0.05), respectively. The median for knowledge of risk factors for DVT attained by nurses >31 years old was significantly higher than the median of nurses aged 26–30 years and <25 years old, (15), (12) and (12) (p<0.05), respectively. The median for knowledge of the prevention of DVT attained by nurses >31 years old was significantly higher than the median of nurses <25 years old, (6) and (5) (p<0.05), respectively.

**Table 3. Nurses knowledge of DVT in groups stratified by demographic characteristics.**

| N (%) | | Nurses practice on prevention score | | Nurses general knowledge score | | Nurses knowledge of risk factor score | | Nurses knowledge of prevention score | |
|---|---|---|---|---|---|---|---|---|---|
| | | Median (IQR) | P | Median (IQR) | p | Median (IQR) | P | Median (IQR) | p |
| **Gender** | | | | | | | | | |
| Males | 72 (31) | 18 (9.75) | >0.05 | 4 (2) | >0.05 | 12 (3) | >0.05 | 5 (2) | >0.05 |
| Females | 160 (69) | 19 (10) | | 4 (2) | | 12 (4) | | 5 (2) | |
| **Age** | | | | | | | | | |
| <25 | 124 (53.4) | 21 (10) | <0.05 | 4 (2) | <0.05 | 12 (4) | <0.05 | 5 (2) | <0.05 |
| 26–30 | 73 (31.5) | 18 (10) | | 4 (2) | | 12 (4) | | 5 (2) | |
| > 31* | 35 (15.1) | 17 (6) | | 5 (2) | | 15 (5) | | 6 (2) | |
| **Experience** | | | | | | | | | |
| 1–5 | 153 (65.9) | 18 (11) | <0.05 | 4 (2) | <0.05 | 12 (4) | >0.05 | 5 (2) | >0.05 |
| 6–10 | 50 (21.6) | 19 (9.5) | | 4 (1.2) | | 11 (4) | | 5 (2) | |
| > 11 | 29 (12.5) | 17 (7.5) | | 5 (3) | | 14 (5.5) | | 6 (2.5) | |
| **Education** | | | | | | | | | |
| Diploma | 65 (28) | 18 (9.5) | >0.05 | 4 (2) | >0.05 | 12 (3) | >0.05 | 5 (2) | >0.05 |
| Bachelor | 136 (58.6) | 19 (10.7) | | 4 (2) | | 12 (4.7) | | 5 (2) | |
| Master | 31 (13.4) | 18 (11) | | 4 (2) | | 14 (4) | | 5 (2) | |
| **Working Unit** | | | | | | | | | |
| Emerg | 37 (15.9) | 18 (6) | <0.05 | 4 (2) | <0.05 | 13(3.5) | <0.05 | 6 (2.5) | >0.05 |
| ICU | 36 (15.5) | 22 (11) | | 3 (1.7) | | 11 (3) | | 5 (2.7) | |
| Internal | 39 (16.8) | 18 (11) | | 4 (2) | | 13 (5) | | 5 (4) | |
| Gynae | 21 (9.1) | 16 (5.5) | | 5 (2.5) | | 14 (6) | | 6 (2) | |
| Onco | 16 (6.9) | 27 (6.7) | | 3 (1.7) | | 12 (3) | | 5 (1.7) | |
| Sugery | 29 (12.5) | 18 (11.5) | | 4 (2) | | 12 (4.5) | | 5 (2) | |
| Polycli | 15 (6.5) | 15 (7) | | 4 (3) | | 14 (5) | | 5 (2) | |
| Orthopaed | 29 (12.5) | 20 (8.5) | | 6 (2) | | 11 (3) | | 6 (2) | |

^ Kruskal-Wallis test and Mann-Whitney U tests were used for the statistical analyses, when applicable. IQR (Interquartile range).

Regarding the number of years of experience, nurses with >11 years of experience had a median for general practice that was significantly lower than the median of the nurses with 6–10 years of experience (17) and (19*) (p<0.05), respectively. Also, no statistically significant differences were observed in the four different scores between education subgroups (p>0.05).

Regarding the work units, nurses who worked in an ICU had a median for practice that was significantly higher than the median of the nurses who worked in gynaecology, (22) and (16) (p<0.05), respectively. The nurses who worked in gynaecology unit attained a median for general knowledge that was significantly higher than the median of the nurses who worked in both polyclinic and an oncology unit, (5) (4) and (3.5) (p<0.05), respectively. The nurses who worked in an ICU had median for risk factor knowledge that was significantly lower than the median of the nurses who worked in polyclinic units, (11) and (14) (p<0.05), respectively.

## Physicians' demographics, knowledge, and attitudes towards thromboprophylaxis

One hundred seventeen questionnaires were dispersed to physicians, and 109 were returned, corresponding to a response rate of approximately 93%. One hundred three questionnaires were evaluated, while 6 that were improperly filled were discarded. Physicians who responded to questionnaires were professors (n = 29), associate professors (n = 15), assistant professors

**Table 4. Descriptive statistics of knowledge and attitude scores of physicians.**

| Variables | Mean | Standard deviation | Minimum | Maximum |
|---|---|---|---|---|
| Knowledge | 6.58 | 2.37 | 0.00 | 11 |
| Attitude | 20.12 | 4.86 | 9.00 | 30 |

(n = 18) and specialists (n = 41) working in different clinics. Table 4 presents the descriptive statistics of knowledge and attitude scores for physicians. Regarding the knowledge of physicians who completely responded to the questionnaire, a high percentage of incorrect answers were observed for most of the questions (10 of 15 questions). More than 50% of physicians did not know that VTE is a fatal combination of DVT. Similarly, 77.7% of physicians did not know that the administration of general anaesthesia for <30 minutes does not increase the risk of deep venous thrombosis. However, the most common correct knowledge answers were recorded for the question "Patients undergoing surgery are more susceptible to deep venous thrombosis/venous thromboembolism than medical patients" (76.7%).

In response to attitude questions, the majority of the respondent (38.8%) stated that they "Strongly Agree" that prevention/prophylaxis of DVT is necessary prior to surgery, and only (16.5%) stated that they "Strongly Disagree" that educating patients regarding preventive measures of DVT is necessary. Furthermore, they indicated a requirement for routine ultrasound screening in asymptomatic patients at discharge or during outpatient follow-up, as shown in Table 5.

## Discussion

Indeed, after assessing 180 patients using the Caprini risk assessment tool, finding of the current study show that thromboprophylaxis regimens were appropriately provided to only approximately 52.3% of patients, consistent with the studies by White RH et al. [20], Nekoonam B et al. [21] and Kingue et al. [22], where 50%, 32.6% and 58.5% of the subjects received correct prophylaxis, respectively. In contrast, 20.3% of patients examined in the study by Cristiano et al. [23] received rational prophylaxis and venous thromboembolism is still the major cause of their sudden death. The results are also consistent with the findings reported by Sharif-Kashani et al. showing that rational prophylaxis was provided to less than half of the patients included in the study [24]. In our study, 3.4% of patients received inadequate prophylaxis (e.g., insufficient doses of enoxaparin or compression stockings alone), in contrast to the

**Table 5. Responses of physicians to questions examining attitudes towards DVT (N = 103).**

| Attitude statements | Strongly Disagree N (%) | Disagree N (%) | Neutral N (%) | Agree N (%) | Strongly Agree N (%) | Mean ± SD | Total attitude score |
|---|---|---|---|---|---|---|---|
| **1. I believe that Doppler sonography (sensitive and objective tests) is necessary to screen for post-surgical DVT in patients.** | 12 (11.7) | 20 (19.4) | 26 (**25.2**) | 30 (29.1) | 15 (14.6) | 3.16 ± 1.2 | 20.12 ± 4.9 |
| **2. I believe that an assessment of DVT risk factors is necessary prior to surgery.** | 13 (12.6) | 14 (13.6) | 17 (16.5) | 26 (25.2) | 33 (**32**) | 3.50 ± 1.39 | |
| **3. I believe that the prevention/prophylaxis of DVT is necessary prior to surgery.** | 7 (6.8) | 19 (18.4) | 17 (16.5) | 20 (19.4) | 40 (**38.8**) | 3.65 ± 1.34 | |
| **4. I believe that educating patients regarding preventive measures of DVT is necessary.** | 17 (16.5) | 16 (15.5) | 22 (21.4) | 34 (**33**) | 14 (13.6) | 3.12 ± 1.30 | |
| **5. I believe that nurses require training in methods to prevent DVT.** | 13 (2.6) | 17 (16.5) | 27 (**26.2**) | 19 (18.4) | 27 (**26.2**) | 3.29 ± 1.33 | |
| **6. I believe that the prevention of DVT with low dose heparin is irrational before surgery.** | 15 (14.6) | 8 (7.8) | 26 (25.2) | 28 (**27.2**) | 26 (25.2) | 3.41 ± 1.33 | |

results by Zeitoun et al. [25] and Nekoonam B et al., where inadequate VTE prophylaxis was administered to 35% and 17.3% of the subjects, respectively. In our study, 15.1% of patients received a higher dose, but 6.52% of patients analysed in the study by Nekoonam B et al. received higher doses for thromboprophylaxis. Compared to risk scores in the study by Nekoonam B et al., 73.08% of all patients had a high risk with a risk score of 3 or more points, 11.5% had a moderate risk with a risk score of 2 points, and 15.3% had a low risk with a risk score of 1 or fewer points. Our study obtained similar results, where 77.8% of patients had a high level of risk, 15% of patients displayed a moderate level of risk, and only 5.6% and 1.7% displayed a low and very low level of risk, respectively.

According to a study conducted in London [26], 16% and 20% [21] of patients treated with enoxaparin required dose adjustments upon administration, while in our study, only 13.9% of patients administered enoxaparin required a dose adjustment. The enoxaparin prescription pattern identified in the present study was inappropriate, similar to the studies by Fahimi et al. and Nekoonam B et al. As shown in the study by Fahimi et al. [27], the improper dosing, administration, and prescription of enoxaparin occur frequently, and health care providers require training programs and the implementation of evidence-based protocols to control prescription patterns. Regarding complications, a study by Novo-Veleiro et al. [28] reported wound haematoma (7.3%) and major bleeding (0.5%) as the main complications, while wound haematoma occurred in 16.6% of patients and no major bleeding was noted during and after hospitalization in our study.

Regarding the knowledge of the prevention of DVT, most of the nurses had a low score of knowledge, similar to the results of a quantitative study conducted by Abin et al., which concluded that 42% of the nurses attained a low score for knowledge of deep venous thrombosis prevention in hospitalized patients [29]. In the present study, most respondents (88.8%) require DVT education, and this issue should be taken into account to improve the awareness and willingness of nurses to attend training programs, workshops and congresses on the prevention of DVT.

Regarding the evaluation of the knowledge and attitudes of physicians, more than 50% of physicians did not know that VTE is a fatal combination of deep venous thrombosis. Similarly, 77.7% of physicians did not know that the administration of general anaesthesia for <30 minutes does not increase the deep venous thrombosis risk, consistent with the result reported by Mehdi et al. showing that more than 50% of the study population recorded a similar answer. In addition, more than half of physicians did not know that surgery posed a higher risk for patients with cancer to develop deep venous thrombosis than in obese or aged patients [17]. The American College of Chest Physicians (ACCP) recommend that patients must be classified as having very high, high, moderate, and low risks of developing VTE, and a prophylaxis method must be used according to this risk stratification score and every health care setting must develop a formal and effective strategy for the prevention and complication of venous thromboembolism [30]. Thromboprophylaxis was underutilized in tertiary care hospitals in Northern Cyprus which denote a gap between evidence-based guidelines and practice. By giving proper training to (HCPs) about DVT prevention and establishing a hospital-wide clinical Pharmacist based DVT prevention program will decrease the morbidity and mortality associated with this disease process and will assure rational practices in North Cyprus.

## Strength and limitations of the study

The present study assesses the rational use of thromboprophylaxis therapy in hospitalized patients and perceptions of health care providers in two tertiary care hospitals in North Cyprus. However, this study also has some limitations that might decrease the generalizability of the results. As only 2 hospitals were chosen as study setting, we may not be able to generalize the study findings overall hospitals in North Cyprus. Both studied hospitals were teaching

hospitals, in which healthcare professionals provide beside complex care; clinical education and training to current and future health professionals through educational and mentoring activities [31]. Teaching hospitals tend to be early adopters of new evidence and technologies which leads to better outcomes and less mortality compared to non-teaching hospitals [32]. Healthcare providers in teaching hospitals are more exposed to learning and teaching activities besides their preceptorship which encourages them to be theoretically and practically prepared for the role, adheres more closely to clinical policies, best practices and deliver high-quality care and services as role models [33]. This may further suggest inferior knowledge and practice of DVT prophylaxis in other settings with less teaching and mentorship, which necessitate further research and comparison to reach such a conclusion.

The demographic data of the physicians were not collected to increase the response rate. we were unable to document pulmonary embolism as the cause of death of the patients who died during hospitalization because it was not documented properly.

An interventional program that incorporates both education and a daily individual assessment of DVT risk factors is needed with an enclosed prophylaxis policy. The establishment of an effective deep venous thrombosis prophylaxis strategy in health care settings with evidence-based recommendations may be useful to improve patient safety, quality of life, and best practices. Clinical pharmacists can utilize the Caprini risk assessment tool and assist health care providers in the rational implementation of the rational use of medications and antithrombotic prophylaxis in hospitals. Investments in training health care providers about deep venous thrombosis prophylaxis are needed to achieve the proper utilization of antithrombotic medications, this public health issue and regular medication errors related to inappropriate anticoagulant use deserve further consideration to decrease morbidity and mortality.

## Conclusions

Based on the findings of the present study and international reports, adherence to VTE prophylaxis is still low in practice, a high level of irrationality in thromboprophylaxis therapy of hospitalized patients, and inappropriate administration of anticoagulants was observed. Furthermore, a low degree of knowledge of risk factors for deep venous thrombosis, preventive measures, bad practices in preventing deep venous thrombosis among nurses and, a lack of knowledge of health care providers and standard guidelines was also noted in assessed hospitals.

## Supporting information

**S1 Appendix.**
(DOCX)

**S2 Appendix.**
(DOCX)

**S3 Appendix.**
(DOCX)

**S4 Appendix.**
(DOCX)

## Acknowledgments

The authors have gratefully acknowledged the Near East University hospital, for the supply and guidance on the subject, Department of Clinical Pharmacy, Faculty of Pharmacy, Near

East University, Nicosia, North Cyprus, for providing necessary facilities. The authors would like to give special thanks to Dr. Wahab Ali Shah, Dr. Louai M Saloumi and Sibel Severler to provide help and statements that greatly improved the manuscript preparation.

## Author Contributions

**Conceptualization:** Syed Sikandar Shah, Abdikarim Abdi.

**Data curation:** Syed Sikandar Shah.

**Formal analysis:** Syed Sikandar Shah.

**Methodology:** Syed Sikandar Shah, Abdikarim Abdi, Barçın Özcem.

**Resources:** Barçın Özcem.

**Supervision:** Bilgen Basgut.

**Writing – original draft:** Syed Sikandar Shah.

**Writing – review & editing:** Abdikarim Abdi, Barçın Özcem.

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
