## [Decision Letter · Decision Letter 0]

19 Mar 2020

PONE-D-20-04896

The rational use of thromboprophylaxis therapy in hospitalized patients and the perspectives of health care providers in Northern Cyprus

PLOS ONE

Dear MR SHAH,

Thank you for submitting your manuscript to PLOS ONE. After careful consideration, we feel that it has merit but does not fully meet PLOS ONE’s publication criteria as it currently stands. Therefore, we invite you to submit a revised version of the manuscript that addresses the points raised during the review process.

We would appreciate receiving your revised manuscript by May 03 2020 11:59PM. To enhance the reproducibility of your results, we recommend that if applicable you deposit your laboratory protocols in protocols.io, where a protocol can be assigned its own identifier (DOI) such that it can be cited independently in the future. For instructions see: http://journals.plos.org/plosone/s/submission-guidelines#loc-laboratory-protocols

We look forward to receiving your revised manuscript.

Kind regards,

Joel Msafiri Francis, MD, MS, PhD

Academic Editor

PLOS ONE

Journal Requirements:

2. Please provide additional details regarding participant consent. In the ethics statement in the Methods and online submission information, please ensure that you have specified (a) whether consent was informed and (b) what type you obtained (for instance, written or verbal, and if verbal, how it was documented and witnessed). If your study included minors, state whether you obtained consent from parents or guardians. If the need for consent was waived by the ethics committee, please include this information.”

3. Please address the following:

a) Please include additional information regarding the survey or questionnaire used in the study and ensure that you have provided sufficient details that others could replicate the analyses. For instance, if you developed a questionnaire as part of this study and it is not under a copyright more restrictive than CC-BY, please include a copy, in both the original language and English, as Supporting Information. Please also include the exact number of individuals involved in the pilot testing of the questionnaire.

b) Please refer to any sample size calculations performed prior to participant recruitment. If these were not performed please justify the reasons. Please refer to our statistical reporting guidelines for assistance (https://journals.plos.org/plosone/s/submission-guidelines.#loc-statistical-reporting).

Reviewers' comments:

Reviewer's Responses to Questions

**Comments to the Author**

1. Is the manuscript technically sound, and do the data support the conclusions?

Reviewer #1: Partly

Reviewer #2: Partly

2. Has the statistical analysis been performed appropriately and rigorously? 

Reviewer #1: No

Reviewer #2: Yes

3. Have the authors made all data underlying the findings in their manuscript fully available?

Reviewer #1: No

Reviewer #2: Yes

4. Is the manuscript presented in an intelligible fashion and written in standard English?

Reviewer #1: Yes

Reviewer #2: Yes

5. Review Comments to the Author

Reviewer #1: Reviewer’s comments

1. What is the aim of the study? There more than one version

a. This study aims to evaluate the current thromboprophylaxis practice and management of patients with risks of developing DVT in different clinics, to determine the adherence to thromboprophylaxis guidelines and to assess healthcare providers’ (HCPs) knowledge, practice and attitudes towards deep vein thrombosis risks and prophylaxis with the goal of optimizing care and ensuring rational practices.

b. An observational study was conducted in which patients from multiple clinics were enrolled to investigate the rational use of DVT prophylaxis using the Caprini risk assessment tool

c. This study aims to investigate current thromboprophylaxis practice at two university hospitals in North Cyprus by evaluating the management of patients with a low, medium, and high risk of developing DVT who are treated in different clinics to determine the adherence to thromboprophylaxis guidelines and to assess healthcare providers’ knowledge, practices and attitudes towards deep vein thrombosis risks and prophylaxis

Please harmonize the statements in different sections

2. Where was the study done?

a. An observational study was conducted in which patients from multiple clinics were enrolled to investigate the rational use……..

b. The study was conducted in the general wards of two leading tertiary university hospitals, namely, Near East University Hospital (NEUH) and Dr. Suat Günsel Kyrenia University Hospital (SGKUH) in

c. Study settings are not described

3. The inclusion criteria were acute and chronically ill medical patients…. But table 1 shows patients from diverse wards – including orthopaedic and surgery? This needs to be described.

4. Although the authors reported having followed up participants to asses for possible complications, the development of post-discharge complications (deep venous thrombosis signs and symptoms, pulmonary embolism or adverse effects of medications) was not obvious in the manuscript.

5. The statement “The average length of hospitalization was 12.48 days (median of 11 days)” is unclear. Did the authors report the mean (average – 12.48) and median (11) at the same time? Why were the two measures reported together?

6. Table 2 describes "major" risk factors. Why are these risk factors called major? I am not sure if the term major is used in the Caprin score.

7. In some areas, absolute numbers and percentages are given without reference to what is the denominator. A reader would struggle to identify the denominator. One of these sentences includes "Four of these patients (30.76%) developed 6 minor complications. Anticoagulation therapy was stopped in 2 patients (50%)”

8. The importance of table 3 is unclear. What does table 3 answer? It looks redundant if there is nothing exclusive that it adds to the study.

9. There are several areas where authors used the word "female" to refer to "females."

Reviewer #2: Reviewer Comments

Manuscript Title: The rational use of thromboprophylaxis in hospitalized patients and the perspectives of health care providers in Northern Cyprus

General Impression: The paper describes an important area of thromboprophylaxis. However, some clarifications to the manuscript are required.

1. Materials and Methods:

a) Study Setting and Subjects: “All inpatients admitted between 01 April 2018 and 01 July 2018 who met the inclusion criteria were included in the analysis”. The authors need to clarify and edit the exclusion criteria stated. (lines 87 to 89) as the language is unclear

b) Did the authors have a predetermined minimum required sample size for their analysis plan?

c) The authors also need to clarify at what point data was initially collected-did this happen while patient was still on the ward or after they had been discharged. Also, it’s not clear what data was collected from the patient’s chart/record

d) Ethical consideration: The authors clearly state that they received IRB ethics approval. However, it is also essential to state whether study participants provided written or verbal informed consent prior to participation and if not, state justification for waiver of this requirement

2. Observational Results:

a) Patient demographics and characteristics:

The authors need to mention the number of patients admitted during the 3-month study period from which they selected participants as stated in the methods section.

Table 1: Table shows that the patients included some from the Orthopedics and Surgery clinics- The authors need to clarify their patient sample description in the Methods Section (line 85)-Are these ‘medical’ patients?

To improve readability, authors should consider making the percentage proportions reported consistent by using uniform number of decimal values throughout the manuscript.

3. Discussion:

a) The authors have made many comparisons of their findings to those from similar studies elsewhere. However, I expected the authors to also expound the implications of their findings on the knowledge gaps in relation to those on rationality of thromboprophylaxis in this study

b) Limitations: Was the sampling method ideal and was the sample size for the different phases adequate to draw conclusions? This needs to be considered.

Both study sites are university (teaching) hospitals. Could this have affected the findings on knowledge levels of physicians/nurses?

Also, whilst the authors assessed nurse and physicians’ knowledge on DVT, it is not clear if the HCWs are aware of the Caprini’s RAM and if this is actually standard of practice. Is it possible that the low utilization/rationalization of thromboprophylaxis could have been due to lack of awareness?

Finally,the authors need to check the language and revise text where necessary throughout the manuscript to improve readability.Ensure the references are correctly written

6. PLOS authors have the option to publish the peer review history of their article (what does this mean?). If published, this will include your full peer review and any attached files.

Reviewer #1: No

Reviewer #2: No

---

## [Author Response · Author response to Decision Letter 0]

3 May 2020

Dear reviewers and academic editor, 

Thanks much for your time and fruitful comments on the article “The rational use of thromboprophylaxis therapy in hospitalized patients and the perspectives of health care providers in Northern Cyprus” Your review and comments would further help enhance the readability and the delivering of the message this article carries.

In response to reviewers section, we reviewed and corrected all the questions raised by the academic editor and reviewers and described the enhancement made on the article following your comments and recommendations which further improved the readability of the manuscript.

Thanks much for your effort and thus contribution to our work.

---

## [Decision Letter · Decision Letter 1]

21 May 2020

PONE-D-20-04896R1

The rational use of thromboprophylaxis therapy in hospitalized patients and the perspectives of health care providers in Northern Cyprus

PLOS ONE

Dear Dr. SHAH,

Thank you for submitting your manuscript to PLOS ONE. After careful consideration, we feel that it has merit but does not fully meet PLOS ONE’s publication criteria as it currently stands. Therefore, we invite you to submit a revised version of the manuscript that addresses the points raised during the review process.

We look forward to receiving your revised manuscript.

Kind regards,

Joel Msafiri Francis, MD, MS, PhD

Academic Editor

PLOS ONE

Reviewers' comments:

Reviewer's Responses to Questions

**Comments to the Author**

1. If the authors have adequately addressed your comments raised in a previous round of review and you feel that this manuscript is now acceptable for publication, you may indicate that here to bypass the “Comments to the Author” section, enter your conflict of interest statement in the “Confidential to Editor” section, and submit your "Accept" recommendation.

Reviewer #1: (No Response)

Reviewer #2: All comments have been addressed

2. Is the manuscript technically sound, and do the data support the conclusions?

Reviewer #1: No

Reviewer #2: Yes

3. Has the statistical analysis been performed appropriately and rigorously? 

Reviewer #1: No

Reviewer #2: Yes

4. Have the authors made all data underlying the findings in their manuscript fully available?

Reviewer #1: No

Reviewer #2: Yes

5. Is the manuscript presented in an intelligible fashion and written in standard English?

Reviewer #1: Yes

Reviewer #2: Yes

6. Review Comments to the Author

Reviewer #1: 1.The sample size calculation is stil not clear. Authors reported using 50% response distribution as one of the parameters in the sample size calculation. What was this response referring to? Is this a justifiable way of sample size calculation?

2.Mean and median still appear in table 3. Is there a need for continous variables to be described/ summarised in both ways?

3. In table 3: How were the comparisons made? were mean or median used for the comparisons of continous variables?

4. The Mann-Whitney U test and the Kruskal-Wallis tests were reported as the methods used to compare diffrences accross groups. One would then assume that the data were deemd as being assymetrical . Yet, mean(and SD) were compared as seen in the texts (line 264, 267, 270, 272. etc)

Reviewer #2: The authors have addressed all my previous comments satisfactorily.

The authors should have a second look at the wording and punctuation and correct any typos in the manuscript

7. PLOS authors have the option to publish the peer review history of their article (what does this mean?). If published, this will include your full peer review and any attached files.

Reviewer #1: Yes: Julius Mwita

Reviewer #2: No

---

## [Author Response · Author response to Decision Letter 1]

4 Jun 2020

Dear reviewers and academic editor, thanks much for your time and fruitful comments on the article “The rational use of thromboprophylaxis therapy in hospitalized patients and the perspectives of health care providers in Northern Cyprus” Your review and comments would further help enhance the readability and the delivering of the message this article carries.

---

## [Decision Letter · Decision Letter 2]

12 Jun 2020

PONE-D-20-04896R2

The rational use of thromboprophylaxis therapy in hospitalized patients and the perspectives of health care providers in Northern Cyprus

PLOS ONE

Dear Dr. SHAH,

Thank you for submitting your manuscript to PLOS ONE. After careful consideration, we feel that it has merit but does not fully meet PLOS ONE’s publication criteria as it currently stands. Therefore, we invite you to submit a revised version of the manuscript that addresses the points raised during the review process.

We look forward to receiving your revised manuscript.

Kind regards,

Joel Msafiri Francis, MD, MS, PhD

Academic Editor

PLOS ONE

Additional Editor Comments (if provided):

Please kindly address the few additional comments from the reviewers.

Reviewers' comments:

Reviewer's Responses to Questions

**Comments to the Author**

1. If the authors have adequately addressed your comments raised in a previous round of review and you feel that this manuscript is now acceptable for publication, you may indicate that here to bypass the “Comments to the Author” section, enter your conflict of interest statement in the “Confidential to Editor” section, and submit your "Accept" recommendation.

Reviewer #1: All comments have been addressed

Reviewer #2: (No Response)

2. Is the manuscript technically sound, and do the data support the conclusions?

Reviewer #1: Yes

Reviewer #2: Partly

3. Has the statistical analysis been performed appropriately and rigorously? 

Reviewer #1: Yes

Reviewer #2: Yes

4. Have the authors made all data underlying the findings in their manuscript fully available?

Reviewer #1: No

Reviewer #2: Yes

5. Is the manuscript presented in an intelligible fashion and written in standard English?

Reviewer #1: Yes

Reviewer #2: Yes

6. Review Comments to the Author

Reviewer #1: All the comments have been addressed. No more comments from my side. The manuscript can be accepted for publication

Reviewer #2: Abstract: Ensure you use a uniform tense throughout the abstract.The 'Aims' section is stated in present tense.

Ensure that the proportions mentioned here match those in the results and discussion sections

Discussion:

Line 313

"Indeed, after assessing 180 patients using the Caprini risk assessment tool, finding of the current study show that thromboprophylaxis regimens were appropriately provided to only approximately 52.2% of patients, consistent with the studies by White RH et al..."

The percentage proportion stated here does not add up with that stated in the abstract (47.7%).The authors should consider correcting this statement to reduce its ambiguity .

Conclusion:

There is some repetition in the authors conclusion.It needs to be made more succinct and focused on their study

7. PLOS authors have the option to publish the peer review history of their article (what does this mean?). If published, this will include your full peer review and any attached files.

Reviewer #1: Yes: Julius Mwita

Reviewer #2: No

---

## [Author Response · Author response to Decision Letter 2]

14 Jun 2020

Dear reviewers and academic editor, thanks much for your time and fruitful comments on the article “The rational use of thromboprophylaxis therapy in hospitalized patients and the perspectives of health care providers in Northern Cyprus” Your review and comments would further help enhance the readability and the delivering of the message this article carries.

---

## [Decision Letter · Decision Letter 3]

17 Jun 2020

The rational use of thromboprophylaxis therapy in hospitalized patients and the perspectives of health care providers in Northern Cyprus

PONE-D-20-04896R3

Dear Dr. SHAH,

We’re pleased to inform you that your manuscript has been judged scientifically suitable for publication and will be formally accepted for publication once it meets all outstanding technical requirements.

Kind regards,

Joel Msafiri Francis, MD, MS, PhD

Academic Editor

PLOS ONE

Additional Editor Comments (optional):

Reviewers' comments:

Reviewer's Responses to Questions

**Comments to the Author**

1. If the authors have adequately addressed your comments raised in a previous round of review and you feel that this manuscript is now acceptable for publication, you may indicate that here to bypass the “Comments to the Author” section, enter your conflict of interest statement in the “Confidential to Editor” section, and submit your "Accept" recommendation.

Reviewer #2: All comments have been addressed

2. Is the manuscript technically sound, and do the data support the conclusions?

Reviewer #2: Yes

3. Has the statistical analysis been performed appropriately and rigorously? 

Reviewer #2: Yes

4. Have the authors made all data underlying the findings in their manuscript fully available?

Reviewer #2: Yes

5. Is the manuscript presented in an intelligible fashion and written in standard English?

Reviewer #2: Yes

6. Review Comments to the Author

Reviewer #2: The authors have addressed all the comments in the 'Response to reviewer comments' and just need to double check that these changes are included in the main text.For example the proportion of rationally managed patients is 52.3% according to figure 1 but is stated as 52.2% in the abstract and discussion sections.

7. PLOS authors have the option to publish the peer review history of their article (what does this mean?). If published, this will include your full peer review and any attached files.

Reviewer #2: No

---

## [Editor Report · Acceptance letter]

22 Jun 2020

PONE-D-20-04896R3 

The rational use of thromboprophylaxis therapy in hospitalized patients and the perspectives of health care providers in Northern Cyprus 

Dear Dr. Shah:

I'm pleased to inform you that your manuscript has been deemed suitable for publication in PLOS ONE. Congratulations! Your manuscript is now with our production department. 

Kind regards, 

on behalf of

Dr. Joel Msafiri Francis 

Academic Editor

PLOS ONE